# Layer-by-Layer Heterostructure of MnO_2_@Reduced Graphene Oxide Composites as High-Performance Electrodes for Supercapacitors

**DOI:** 10.3390/membranes12111044

**Published:** 2022-10-26

**Authors:** Tingting Liu, Lei Chen, Ling Chen, Guoxing Tian, Mingtong Ji, Shuai Zhou

**Affiliations:** 1Qinhuangdao Key Laboratory of Marine Oil and Gas Resource Exploitation and Pollution Prevention, Northeast Petroleum University at Qinhuangdao, Qinhuangdao 066004, China; 2Provincial Key Laboratory of Polyolefin New Materials, College of Chemistry & Chemical Engineering, Northeast Petroleum University, Daqing 163318, China; 3Hebei Key Laboratory of Applied Chemistry, College of Environmental and Chemical Engineering, Yanshan University, Qinhuangdao 066004, China

**Keywords:** MnO_2_, reduced graphene oxide (rGO), layer-by-layer, supercapacitor

## Abstract

In this paper, *δ*-MnO_2_ with layered structure was prepared by a facile liquid phase method, and exfoliated MnO_2_ nanosheet (e-MnO_2_) was obtained by ultrasonic exfoliation, whose surface was negatively charged. Then, positive charges were grafted on the surface of MnO_2_ nanosheets with a polycation electrolyte of polydiallyl dimethylammonium chloride (PDDA) in different concentrations. A series of e-MnO_2_@reduced graphene oxide (rGO) composites were obtained by electrostatic self-assembly combined with hydrothermal chemical reduction. When PDDA was adjusted to 0.75 g/L, the thickness of e-MnO_2_ was ~1.2 nm, and the nanosheets were uniformly adsorbed on the surface of graphene, which shows layer-by-layer morphology with a specific surface area of ~154 m^2^/g. On account of the unique heterostructure, the composite exhibits good electrochemical performance as supercapacitor electrodes. The specific capacitance of e-MnO_2_-0.75@rGO can reach 456 F/g at a current density of 1 A/g in KOH electrolyte, which still remains 201 F/g at 10 A/g. In addition, the capacitance retention is 98.7% after 10000 charge-discharge cycles at 20 A/g. Furthermore, an asymmetric supercapacitor (ASC) device of e-MnO_2_-0.75@rGO//graphene hydrogel (GH) was assembled, of which the specific capacitance achieves 94 F/g (1 A/g) and the cycle stability is excellent, with a retention rate of 99.3% over 10000 cycles (20 A/g).

## 1. Introduction

Under the background of rapid global economic development, continuous consumption of fossil energy, and increasingly serious environmental pollution, seeking “green” and renewable energy has become the most urgent challenge in society nowadays. Moreover, the breakthrough and popularization of large-scale energy storage technology is strong support for the development of renewable energy. In many forms of energy storage, electrochemical energy storage (EES) has been highly focused on because of its high theoretical conversion efficiency of chemical energy to electrical energy, as well as the high energy density and power density. Accordingly, further technological innovation requires continuous improvement of performance, which also drives the researchers’ exploration of new materials and mechanisms. Compared to batteries, supercapacitors can offer higher power densities and have longer cycle life, faster charge-discharge capabilities and better safety. Therefore, the application of supercapacitors in the field of energy storage has attracted wide attention [1,2].

According to the mechanism of charge storage, supercapacitors can be divided into electrical double-layer capacitors (EDLC) and pseudocapacitors. The electrode materials of EDLC are mainly based on carbon materials, such as activated carbon (AC) [3], carbon nanotube (CNTs) [4,5,6], and graphene [7,8]. For pseudocapacitors, a variety of different materials, such as metal oxides [9,10,11] or hydroxides [12,13], conductive polymers [14,15] and metal sulfides [16], are all candidates for electrode materials. As the first pseudocapacitance electrode material, RuO_2_ has excellent electrochemical performance [17]. However, the toxicity and high cost limit its large-scale application [18]. Some transition metal oxides with low cost also exhibit pseudocapacitive behaviors and can be used in supercapacitors instead of RuO_2_, such as MnO_2_. However, due to poor electronic conductivity, charge storage is limited to a thin layer on the surface, resulting in much lower actual capacitance than its theoretical value. In addition, low charge transfer kinetics and slow ion diffusion also affect the rate performance [19,20]. The effective strategies to improve the electrochemical performance are to increase the specific surface area of the materials by designing various nanostructures to increase the active sites or to construct hierarchical porous structures to improve mass transfer [18]. When the MnO_2_ electrode is processed into ultra-thin film, the specific surface area is greatly increased and the specific capacitance can reach more than 1000 F·g^−1^ [21]. Lang et al. [22] proposed a composite structure of nanoporous gold (NPG) with nanocrystalline MnO_2_. NPG allows electrons to pass through MnO_2_ and promotes rapid ion diffusion between MnO_2_ and electrolyte, thus obtaining a high specific capacitance of ~1145 F·g^−1^, which is very close to the theoretical value.

Recombination and doping are representative approaches to improving the electrochemical performance of MnO_2_. The recombination could produce a synergistic effect; thus, the properties of the composite will be much better than that of the single component. The research priority is mainly to load or grow MnO_2_ on porous carbon materials with large surface areas or metal substrates with good conductivity, such as graphene [23,24,25], carbon nanotubes [26], carbon fiber [27], wood-derived carbon (WC) [28], Ag [29] and Ni [30]. Especially for graphene, it is popular in composites, which could improve the electrical conductivity of the composite and reduce the solution resistance [31]. Composite nanostructure also provides an interconnected pathway for electron transportation and electrolyte diffusion [32,33], as well as inhibits the agglomeration of the individual components [34]. In addition, the heterojunction could be reasonably designed and constructed to adjust the electron structure and improve the rate of ion transport and electron transfer [35]. Metal doping (Au, Ag, Co, Al and Na) can also improve the inherent conductivity of MnO_2_ and promote the electrochemical reaction, which is mainly based on the adjustment of the electronic structure [36]. Zong et al. [37] produced a positive electrode of Na-doped MnO_2_ nanosheets@carbon nanotube fibers (CNTFs) with high performance. The thin nanosheets afford a large surface area for the electrode, as well as inserting Na^+^ into MnO_2_ improves the conductivity to deliver a large specific capacitance (743.3 mF·cm^−2^), leading to a broad potential window extended up to 0–1.2 V.

Chen et al. [38] successfully prepared a novel petal-like MnO_2_ nanosheet@carbon sphere (CS) core-shell structure by in situ growth of MnO_2_ on the surface of the carbon sphere by adjusting the amount of KMnO_4_ precursor. Porous carbon spheres have a high specific surface area and suitable pore size distribution, which are suitable for energy storage and electrolyte conversion. In 1 M Na_2_SO_4_ electrolyte, it has a specific capacitance of 231 F·g^−1^ at a current density of 0.5 A·g^−1^ and good cycle stability. The excellent electrochemical performance is due to the unique core-shell structure and the synergistic effect between MnO_2_ and carbon spheres. Ma et al. [35] prepared layered *α*-MnO_2_ nanowire@ultrathin *δ*-MnO_2_ nanosheet core-shell nanostructure by a simple liquid phase technique. The novel hierarchical nanostructure is composed of ultrathin *δ*-MnO_2_ nanosheets grown on the surface of the α-MnO_2_ ultralong nanowire. When the discharge current density is as high as 20 A·g^−1^, the initial specific capacitance of the composite reaches 153.8 F·g^−1^, and the stability remains at 98.1% after 10,000 charge-discharge cycles. The good rate performance and stability of the composite are attributed to the structural characteristics of the two MnO_2_ crystals. A 1D *α*-MnO_2_ nanowire as the core provided a stable skeleton structure, and ultra-thin 2D *δ*-MnO_2_ nanosheets as the shell formed more active sites. Therefore, the synergistic effect of different dimensions is of great benefit to the improvement of the electrochemical performance. In conclusion, constructing special composite microstructure by compounding materials with different properties is considered to be an effective way to obtain excellent performance as supercapacitor electrodes on account of their good synergistic effects, including material synergistic effect, dimensional synergistic effect and heterostructural synergistic effect. Researchers have carried out extensive exploration of the controllable synthesis, morphology control, structural design and electrochemical performance improvement of the composites of MnO_2_ with different dimensional carbon materials [39,40].

In general, MnO_2_ with relatively poor conductivity is coated on the surface of a conductive substrate, such as graphene, which makes the conductive substrate materials unable to contact each other directly, resulting in the increase of the contact resistance between the particles, which seriously affects the electrochemical performance of the composite. At present, the MnO_2_ layer on the surface of the conductive substrate is often very thin. If the thickness increases, the electrochemical performance of the composite will decrease significantly, but the thin oxide layer will reduce the available active material, which is not beneficial to the full play of its excellent electrochemical properties. Together with the low specific capacitance of carbon materials, it is important to optimize the ratio and the structure of carbon and MnO_2_ in the composites and design an MnO_2_/carbon interface. In addition, constructing a layer-by-layer structure is a viable choice. Research on layer-by-layer structure are mostly focused on multilayer films by layer-by-layer self-assembly in the presence of substrates, such as carbon cloth [41], indium-tin-oxide (ITO) [42], Ni foam [43] and gold-coated poly-(ethylene terephthalate) (PET) [44].

In this work, on the basis of the preparation of *δ*-MnO_2_ with a layered structure, MnO_2_ laminas were obtained by ultrasonic exfoliation, which was positively charged after surface charge modification by polydiene dimethyl ammonium chloride (PDDA). As a commonly used cationic polyelectrolyte, PDDA has many advantages, such as safety, non-toxicity, easy solubility in water, strong cohesion, good hydrolysis stability and low cost. There is also another positively charged polyelectrolyte always used for charge regulation, such as polyethyleneimine (PEI). As a strong polyelectrolyte, the electrostatic interaction between PDDA and rGO is stronger than that produced by PEI [45], so PDDA was chosen in our research. Then, the composites with the 2D layer-by-layer structure were acquired with no substrate by the self-assembly of MnO_2_ laminas with the surface positively charged and graphene oxide (GO) nanosheets with the surface negatively charged through electrostatic attraction. The final product, MnO_2_/reduced graphene oxide (rGO), was obtained by the reduction of GO to rGO with glucose as the reductant under mild conditions. As expected, rGO in the layer-by-layer structure has acted as a conductive layer and bridge to improve the electrical conductivity of the MnO_2_/rGO composite and relieved the stacking of MnO_2_ nanosheets. A series of self-assembled MnO_2_/rGO composites were prepared by adjusting the concentration of cationic polymer for surface charge modification. As supercapacitor electrodes, the composite designed with layer-by-layer heterostructure shows high performance.

## 2. Experimental Section

### 2.1. Reagents and Materials

All the reagents used in this part were of analytical grade. KMnO_4_ was purchased from Kemiou Chemical Reagent Co., Ltd. (Tianjin, China). Sodium dodecyl sulfate (SDS) was purchased from Aibi Chemical Reagent Co., Ltd. (Shanghai, China). H_2_SO_4_ (98 wt.%) and HCl (37 wt.%) were purchased from Xilong Scientific Co., Ltd. (Shantou, China). PDDA (20 wt.%) was purchased from Aladdin Reagent Co., Ltd. (Shanghai, China). Glucose was purchased from Damao Chemical Reagent Factory (Tianjin, China). Natural graphite (3000 mesh) was purchased from Huatai Lubrication Seal Technology Co., Ltd. (Qingdao, China). Nickel foam was purchased from Liyuan New Material Co., Ltd. (Changsha, China).

### 2.2. Preparation of the Composites

32 mL of SDS (0.4 mol/L) and 1.6 mL H_2_SO_4_ (0.4 mol/L) were mixed into 283.2 mL deionized water and heated to 95 °C for 15 min under continuous stirring. Then 3.2 mL of KMnO_4_ (0.2 mol/L) solution was added and stirred for 60 min at 95 °C [46]. The product was cooled to room temperature and then centrifuged at 5000 rpm. The centrifuged product was freeze-dried after washing, which was MnO_2_.

400 mg of MnO_2_ above prepared was dispersed in 250 mL deionized water to be exfoliated under ultrasonication (40 KHz, 240 W) for 3.5 h. Every 0.5 h, a stirring of 5 min was needed, and the ice bath environment was always maintained during the ultrasonication process. After ultrasonication, the solution was centrifuged at 5800 rpm, and the upper liquid was freeze-dried to obtain the exfoliated MnO_2_ nanosheets, denoted as e-MnO_2_. The exfoliated MnO_2_ dispersion presents Tyndall effect through testing.

The tested Zeta potential of e-MnO_2_ was −20.2 mV. Positive charge was grafted onto the surface of e-MnO_2_ by PDDA. 80 mg of e-MnO_2_ was dispersed, respectively, in 40 mL of PDDA solution with different concentrations (0.5, 0.75 and 1 g/L) to obtain corresponding e-MnO_2_ assembly solution, wherein the concentration of e-MnO_2_ is all 2 mg/mL. The above e-MnO_2_ positively charged by different concentrations of PDDA were named e-MnO_2_-0.5, e-MnO_2_-0.75 and e-MnO_2_-1, respectively. The Zeta potential of these e-MnO_2_ samples after charge regulation is shown in Appendix A. It can be seen that when the concentration of PDDA is 0.5 g/L, the Zeta potential of e-MnO_2_-0.5 reaches 27 mV. When the concentration increases to 0.75 g/L, the Zeta potential of e-MnO_2_-0.75 increases to 35.7 mV, whereas the concentration of PDDA increases to 1 g/L, the Zeta potential of e-MnO_2_-1 decreases to 30.2 mV. It is mainly because when the amount of PDDA is low, it cannot effectively prevent the coagulation effect of electrolytes on the sol system, but when the amount of PDDA is too high, it will affect the amount of charge in the diffusion layer of the micelle and also cause the decrease of the Zeta potential of the colloid system, which is not beneficial to the stable existence of the sol [47].

GO was prepared using the modified Hummers method [48]. 40 mg of GO was dispersed in 40 mL of deionized water to obtain GO assembly solution (1 mg/mL). The tested Zeta potential of GO is −50 mV. As known, when the absolute value of the potential exceeds 30 mV, stable dispersion can be formed [49]. Under continuous stirring, GO assembly solution was added into e-MnO_2_-0.75 assembly solution slowly, also into e-MnO_2_-0.5 and e-MnO_2_-1 dispersion for comparison, and then stirred for 40 min. During the process, it could be observed that with the addition of GO, coagulation occurred. Positively charged e-MnO_2_-0.75 and negatively charged GO completed electrostatic self-assembly. After static settlement, the supernatant was removed, and then the product was filtrated and freeze-dried, denoted as e-MnO_2_-0.5@GO, e-MnO_2_-0.75@GO and e-MnO_2_-1@GO, respectively. The prepared e-MnO_2_-0.75@GO was dispersed in 40 mL of deionized water. Under magnetic stirring, 100 µL NH_3_·H_2_O solution (25% *w*/*w*) was added to adjust pH to ~9–10, then 640 mg glucose was added. After stirring for 15 min, the mixture was transferred into a Teflon-lined stainless steel autoclave of 50 mL and reacted at 95 °C for 1.5 h in order to reduce GO to rGO. The precipitate was repeatedly washed and freeze-dried to acquire e-MnO_2_-0.75@rGO composite. In the same way, e-MnO_2_-0.5@rGO and e-MnO_2_-1@rGO were synthesized. The schematic synthesis procedure for the e-MnO_2_@rGO composites is illustrated in Figure 1.

### 2.3. Characterization

The phase structures of the as-prepared materials were performed using the powder X-ray diffraction (XRD, Rigaku D-max-2500/PC) with Cu Kα radiation (*λ* = 0.15406 nm) over a range 2*θ* = 5–70°. The elemental analysis was detected by X-ray photoelectron spectroscopy (XPS, Thermo Fischer, ESCALAB Xi+) with Al Kα radiation (h*γ* = 1486.6 eV). The analysis of chemical bond was completed via Fourier transform infrared spectroscopy (FTIR, Thermo Nicolet iS10), and the preparation process of samples for FTIR is detailed in Appendix A. The morphologies were characterized by scanning electron microscopy (SEM, Zeiss Supra55 and Hitachi Regulus SU8230), transmission electron microscopy (TEM, Hitachi HT-7700) and atomic force microscopy (AFM, Bruker Multimode 8). The X-ray energy disperse spectra (EDS) of the samples were recorded on Oxford Instruments (Ultim Max170). The specific surface area was measured by Brunauer-Emmett-Teller (BET) method (Micromeritics Tristar Ⅱ 3020). 

### 2.4. Electrochemical Measurements

The as-prepared material was evenly mixed with acetylene black and polytetrafluoroethylene (PTFE, 60 wt.%) at a mass ratio of 80:15:5, and then a modicum of anhydrous ethanol was added to make a paste, which was smeared on the surface of nickel foam, and dried at 60 °C. The nickel foam loaded with active material was pressed under a pressure of 10 MPa, and then soaked in 6 M KOH solution for 24 h for activation. The electrochemical performance was tested in a three-electrode system with 6M KOH aqueous as electrolyte. The as-prepared active material was used as the working electrode, platinum plate electrode as the counter electrode, and Hg/HgO electrode as the reference electrode. Galvanostatic charge-discharge (GCD) tests were recorded on a charge-discharge instrument (Neware CT-4008T, Shenzhen, China) with a potential range of approximately −0.2–0.5 V. Cyclic voltammetry (CV) and electrochemical impedance spectroscopy (EIS) were carried out on a CHI660E electrochemical workstation (Chenhua, Shanghai, China). The potential window of CV tests was from −0.2 V to 0.5 V, and the frequency range of EIS was from 10^−2^ Hz to 10^5^ Hz with the amplitude of 5 mV.

The asymmetric supercapacitor (ASC) device was assembled by using the as-prepared e-MnO_2_-0.75@rGO as the positive electrode, graphene hydrogel (GH) as the negative electrode and 6 M KOH as electrolyte. The preparation of GH is described in Appendix A. Wherein the mass ratio of positive and negative electrodes was obtained by the equation below [50]:(1)$$\frac{m^{+}}{m^{\;-}}=\frac{C_{s}^{\;-}\;\Delta V^{\;-}}{C_{s}^{+}\;\Delta V^{+}}$$
where *m* is the mass of active materials (g), *C_s_* is the specific capacitance of electrodes (F·g^−1^) and ∆*V* is the potential window (*V*), and the sign of “+” and “−” represents the positive and negative electrodes, respectively.

## 3. Results and Discussion

### 3.1. Structure and Morphology

XRD patterns of MnO_2_, e-MnO_2_, e-MnO_2_-0.5@rGO, e-MnO_2_-0.75@rGO and e-MnO_2_-1@rGO are shown in Figure 2a. For MnO_2_, the diffraction peaks of 2*θ* at 12.3°, 24.9°, 37° and 65.5° can be indexed to birnessite-type MnO_2_ (*δ*-MnO_2_) (PDF# 43-1456), corresponding to (001), (002), (200) and (020) crystal planes, respectively [51]. Simultaneously, rGO was prepared using the same reduction process of the composite. Moreover, XRD patterns of rGO and GO are displayed in Appendix A. The XRD curve of GO has a strong diffraction peak at 2*θ* = 11.8°, corresponding to (001) crystal plane. After reduced by hydrothermal reduction with glucose, a broad diffraction peak (002) appears at 2*θ* = 24.5°, indicating the reduction of GO to rGO. The position of which is close to the (002) plane of *δ*-MnO_2_, so there may be an overlap of (002) peaks for MnO_2_ and rGO [52]. Moreover, the existence of rGO in the composite will be further proved by subsequent SEM, TEM and XPS characterization. According to the Bragg equation, the basal plane spacing calculated from the (001) plane is about 0.72 nm. Compared with MnO_2_, there is no obvious change for the position of the diffraction peaks of e-MnO_2_, indicating that the phase structure of MnO_2_ has no change after ultrasonic exfoliation, but the intensity of the diffraction peaks weakened, especially the peaks corresponding to (001) and (002) planes. For the e-MnO_2_@rGO composites with different concentrations of PDDA (0.5, 0.75 and 1 g/L), they all present the diffraction peaks of *δ*-MnO_2_ only, which is probably because MnO_2_ laminas covered on the surface of rGO [53]. Moreover, with the change of the concentration, the XRD patterns of e-MnO_2_-0.5@rGO, e-MnO_2_-0.75@rGO and e-MnO_2_-1@rGO have little change, suggesting no effect on the phase structure of the composites for the charge regulation on the surface of e-MnO_2_. FTIR spectra were used to further represent the structure of MnO_2_, e-MnO_2_ and e-MnO_2_-0.75@rGO, as shown in Figure 2b. The position of the characteristic bands of MnO_2_, e-MnO_2_ and e-MnO_2_-0.75@rGO is basically similar. The band at 3345 cm^−1^ corresponds to the stretching vibration of the O-H bond of interlayer H_2_O, and the band at 1632 cm^−1^ is assigned to the stretching vibration of the H-O-H bond of bound water. The band at 482 cm^−1^ is attributed to the stretching vibration of the Mn-O bond [54]. The intensity of the band of e-MnO_2_ is a little stronger than MnO_2_, mainly because more functional groups are exposed on the surface of the nanosheets.

The chemical composition and oxidation state of e-MnO_2_-0.75@rGO were conducted by XPS, as shown in Figure 3. The existence of C, O and Mn elements is proved in the e-MnO_2_-0.75@rGO composite (Figure 3a). Figure 3b shows C 1s core-level XPS spectrum, where the peaks located at 284.8, 286.8 and 288.6 eV are assigned to C-C, C-O and O-C=O bonds, respectively [55,56]. The spectrum of O 1s region (Figure 3c) could be deconvolved into three peaks centered at 533.2, 531.7 and 529.4 eV, corresponding to C-O-H, H-O-H and Mn-O bonds, respectively [57,58]. Wherein H-O-H and C-O-H bonds are attributed to the adsorbed water molecules and surface functional groups of rGO in the composite, respectively, and the Mn-O bond belongs to MnO_2_. For Mn 2p core-level, it could be fitted into four peaks at 654.8 eV, 652.9 eV, 644.4 eV and 641.8 eV (Figure 3d), which are assigned to Mn^4+^(2p_1/2_), Mn^3+^(2p_1/2_), Mn^4+^(2p_3/2_) and Mn^3+^(2p_1/2_), respectively [59,60]. The existence of Mn^3+^ is probably to maintain charge neutrality and oxygen vacancies in the MnO_2_ lattice [60]. The XPS results further confirm that it has been synthesized successfully of the e-MnO_2_-0.75@rGO composite.

The SEM image of the as-prepared MnO_2_ is shown in Figure 4a. It exhibits a large area and continuous lamellar morphology with the size of several hundred nanometers, and the nanosheets cross with each other. AFM was used to represent the morphology of e-MnO_2_. As shown in Figure 4b, smooth nanosheets are observed, which have a large surface with a dimension of ~800 nm from the AFM image (Figure 4b, on the left), and a thickness of ~1.2 nm from the height profile (Figure 4b, on the right). The theoretical thickness of the single-layer MnO_2_ nanosheet is 0.52 nm [42], but it should be considered the existence of hydration on both sides of the single-layer nanosheet [61]. So the thickness of the obtained e-MnO_2_ nanosheet is approximately two layers. The morphologies of e-MnO_2_@rGO composites under different concentrations of PDDA were characterized by SEM. When the concentration of PDDA is 0.5 g/L, the nanosheets of e-MnO_2_-0.5@rGO are stacked, and the structure is relatively compact (Figure 4c), indicating that the dispersed lamellas failed to attract effectively and aggregated again. Although rGO could increase the conductivity of the composite, the compact structure makes it difficult for the electrolyte to enter the material. When the concentration of PDDA is 0.75 g/L, it can be observed from Figure 4d that e-MnO_2_-0.75@rGO composite displays a layer-by-layer structure with relatively uniform compounding, demonstrating that rGO and e-MnO_2_ were assembled well. As can be seen from the TEM image of e-MnO_2_-0.75@rGO (Figure 4f), e-MnO_2_ lamellas are distributed on the surface of rGO nanosheets. These e-MnO_2_ lamellas interlace with each other, and the material shows a relatively transparent state, indicating a less-layer structure. For the novel layer-by-layer heterostructure, it has many advantages: Firstly, the compact combination of e-MnO_2_ and rGO not only improves the conductivity of the composite but also mitigates the powder dropping caused by the expansion of MnO_2_ during the charge-discharge process. Furthermore, the cross-linking between e-MnO_2_ nanosheets results in a large number of pores, which make ions in electrolytes easily accessible to the layers of the nanosheets, thus greatly increasing active sites. When the concentration of PDDA increased to 1 g/L, e-MnO_2_-1@rGO presented a lamellar accumulation structure (Figure 4e), indicating that e-MnO_2_ and rGO have not formed a good assembly. As shown in Figure 4g of SEM mapping images, Mn, O and C elements distribute homogeneously over the e-MnO_2_-0.75@rGO architecture, which further proves the existence and interlacing distribution of MnO_2_ and rGO in the composite.

The specific surface area and pore structure of the e-MnO_2_-0.75@rGO composite were investigated by analyzing N_2_ adsorption-desorption isotherms and the pore size distribution curve, as shown in Figure 5. It shows a type IV isotherm with a hysteresis loop at the relative pressure of ~0.6-1.0, indicating the existence of mesoporous structure in e-MnO_2_-0.75@rGO (Figure 5a) [62]. The specific surface area of e-MnO_2_-0.75@rGO is 154.3 m^2^/g, which is higher or comparable compared with that reported in the literature, such as MnO_2_ nanowires/rGO (139.9 m^2^/g) [62], reduced graphene/MnO_2_ (120.2 m^2^/g) [63] and high-reduced graphene (HRGO)/MnO_2_ (159.1 m^2^/g) [63]. For a comparison, adsorption-desorption isotherms of MnO_2_ and rGO are displayed in Appendix A. The specific surface area of MnO_2_ and rGO is 78.7 and 207.8 m^2^/g, respectively. It can be seen that the higher specific surface area of rGO plays a positive role in the composite, and the specific surface area of e-MnO_2_-0.75@rGO composite is significantly improved compared with that of pristine MnO_2_. Additionally, the pore size distribution curve reveals that the average pore size calculated by Barrett-Joyner-Halenda (BJH) model is concentrated from 20 to 40 nm (Figure 5b), demonstrating the mesoporous structure of the as-prepared composite. While the pore size distribution of MnO_2_ and rGO is both concentrated at ~3-4 nm (Appendix A), so the porosity is owing to the composite layer-by-layer architecture formed by MnO_2_ and rGO, which is ascribed to the cross-linked structure constructed by self-assembly of the folded nanosheets. High specific surface area and suitable pore size are favorable for ions transport during the charge-discharge process.

### 3.2. Electrochemical Performance

For the evaluation of electrochemical performance, CV curves at various scan rates and GCD curves at various current densities were tested, as shown in Figure 6. Figure 6a–c show CV curves of e-MnO_2_-0.5@rGO, e-MnO_2_-0.75@rGO and e-MnO_2_-1@rGO, respectively. CV curves of e-MnO_2_@rGO obtained at different PDDA concentrations are similar in shape and approximate in rectangle, indicating that the materials present good pseudocapacitance characteristics. At the same scan rate, the CV curve of e-MnO_2_-0.75@rGO has the maximum current response, demonstrating the maximum specific capacitance. Moreover, GCD curves of e-MnO_2_-0.5@rGO, e-MnO_2_-0.75@rGO and e-MnO_2_-1@rGO are shown in Figure 6d–f), respectively. These GCD curves at different current densities are close to symmetric triangles, indicating good reversibility. At the same current density, the GCD curve of e-MnO_2_-0.75@rGO has the longest discharge time, also manifesting the highest specific capacitance, which is consistent with the CV results. At a current density of 1 A/g, the specific capacitance of e-MnO_2_-0.5@rGO, e-MnO_2_-0.75@rGO and e-MnO_2_-1@rGO calculated by the GCD curves is 236 F/g, 456 F/g and 298 F/g, respectively. Combined with the above Zeta potential results after surface charge regulation, the higher the absolute value of Zeta potential is, the more stable the system is, so the self-assembly effect is better. When the concentration of PDDA was 0.5 g/L, due to the low concentration of the charge regulating solution, the self-assembly via electrostatic gravity was not ideal, and the nanosheets were aggregated and stacked, which reduced the number of the active sites of MnO_2_ and affected its specific capacitance. However, Zeta potential at 1 g/L of PDDA decreased compared with that at 0.75 g/L of PDDA, which did not achieve complete assembly. Furthermore, the higher concentration of PDDA made the long molecular chain at the outer end cause micelle expansion and sedimentation. While excess PDDA polymers mixed in the composite slowed down the agglomeration, it was bound to reduce the electrical conductivity of the material, thus affecting the electrochemical performance, resulting in the decline of the specific capacitance. The results correspond with SEM analysis.

For a comparison, CV curves at 5 mV/s and GCD curves at 1 A/g of MnO_2_, e-MnO_2_ and e-MnO_2_-0.75@rGO are displayed in Figure 7a,b). The CV curve area of e-MnO_2_ is larger than that of pristine MnO_2_. The composite of e-MnO_2_-0.75@rGO shows the largest closed area, indicating the highest specific capacitance among the three. The specific capacitance of MnO_2_, e-MnO_2_ and e-MnO_2_-0.75@rGO is 268, 360 and 456 F/g, respectively, at a current density of 1 A/g, calculated from the charge-discharge curve in Figure 7b. (The average values and standard deviations were given in Appendix A.) Compared with pristine MnO_2_, e-MnO_2_ has a larger specific surface, and more active sites are exposed to the electrolyte, so it has higher specific capacitance than pristine MnO_2_. Furthermore, e-MnO_2_-0.75@rGO composite has the highest specific capacitance, 70% higher than that of pristine MnO_2_, mainly because rGO and e-MnO_2_ were laminated in a layer-by-layer structure, which improves the conductivity of the material and inhibits the stacking of the nanosheets. According to Figure 7c, for the Nyquist plots of EIS, the intercept at the real axis represents the equivalent series resistance (*R_s_*), including electrolyte resistance, contact resistance between the electrode material and the current collector, or the internal resistance of the material [64]. The diameter of the semi-arc intersecting at the real axis represents the charge transfer resistance (*R_ct_*), the value of which is proportional to the *R_ct_*. As can be seen in Figure 7c, the diameter of the semi-arc for e-MnO_2_-0.75@rGO is very small, indicating lower *R_ct_*. The plots were fitted according to the equivalent circuit given in Figure 7d, and the fitting results are also shown in Figure 7d. The *R_s_* of MnO_2_ and e-MnO_2_-0.75@rGO is 0.69 Ω and 0.46 Ω, respectively. Moreover, the corresponding *R_ct_* is 0.42 Ω and 0.26 Ω, manifesting that the *R_s_* and *R_ct_* of e-MnO_2_-0.75@rGO are both lower than pristine MnO_2_. It is further demonstrated that e-MnO_2_-0.75@rGO composite with lamellar structure has good conductivity, and the lamellar structure could contribute to the transfer of electrolyte ions, while rGO is also conducive to reducing the internal resistance of the material. Figure 7e shows the rate performance of MnO_2_, e-MnO_2_-0.5@rGO, e-MnO_2_-0.75@rGO and e-MnO_2_-1@rGO at the current densities increasing from 1 to 10 A/g. Obviously, the capacitance retention of e-MnO_2_-0.5@rGO, e-MnO_2_-0.75@rGO and e-MnO_2_-1@rGO are 16.9%, 44% and 34.6%, respectively, which is more superior than that of pristine MnO_2_ (10.6%), owing to the role of rGO in stabilizing the architecture. In particular, the rate performance of e-MnO_2_-0.75@rGO is significantly the best in the e-MnO_2_@rGO composites with different concentrations of PDDA, and it has the highest specific capacitance at different current densities of 1, 2, 5, 8 and 10 A/g. Figure 7f shows the cycle performance and coulombic efficiency of the e-MnO_2_-0.75@rGO composite. It can be seen that the specific capacitance retention remains at 98.7% of the initial value after 10,000 charge-discharge cycles at a current density of 20 A/g, and coulombic efficiency is about 98.7%.

The specific capacitance of e-MnO_2_-0.75@rGO composite synthesized by electrostatic self-assembly combined with hydrothermal reduction after exfoliating MnO_2_ nanosheets in this work was compared with the composites of MnO_2_ with graphene prepared via a variety of methods reported in the literature [47,63,64,65,66,67,68,69]. As shown in Table 1, in terms of the specific capacitance, the result of our work is comparable.

Compared with pristine MnO_2_, the improved electrochemical performance of e-MnO_2_-0.75@rGO composite is mainly due to the following reasons. Briefly, a layer-by-layer heterostructure of e-MnO_2_-0.75@rGO facilitates the diffusion of electrolyte ions. Meanwhile, the architecture was developed by the self-assembly of graphene nanosheets and e-MnO_2_ nanoflakes, which could effectively inhibit the restacking of e-MnO_2_ and graphene. In addition, the intersecting e-MnO_2_ nanosheets formed a large number of pores, promoting rapid faradaic reactions. Furthermore, the synergistic effect of components must be mentioned. As a conductive layer in the heterostructure, rGO improves the electrical conductivity of the e-MnO_2_-0.75@rGO composite, which is beneficial to the improvement of the overall electrochemical performance [70,71].

Additionally, an asymmetric supercapacitor (ASC) device was assembled, and its electrochemical performance was evaluated. The as-prepared e-MnO_2_-0.75@rGO was used as the positive electrode, graphene hydrogel (GH) was used as the negative electrode and 6 M KOH was used as the electrolyte. Figure 8a shows CV curves of the GH negative electrode and e-MnO_2_-0.75@rGO positive electrode at the scan rate of 5 mV/s. The potential window of the negative electrode is approximately −1–0 V, and that of the positive electrode is approximately −0.2–0.5 V. Figure 8b displays CV curves under various potential windows. It can be observed that CV curves remain a good rectangle when the voltage windows are ~0–1.5 V and ~0–1.6 V; while the voltage rises to 1.7 V, slight polarization occurs. Therefore, ~0–1.6 V was selected as the potential window of the ASC device. GCD curves of e-MnO_2_-0.75@rGO//GH ASC at current densities of 1, 2, 5, 8 and 10 A/g with the potential window of ~0–1.6 V are shown in Figure 8c. All the GCD curves display approximately symmetrical triangles, indicating good reversibility of the device. The calculated specific capacitance could reach 94 F/g at 1 A/g and still keep 35 F/g at 10 A/g. The cycle performance of the device is revealed in Figure 8d. The specific capacitance retention of e-MnO_2_-0.75@rGO//GH ASC can reach 99.3% after 10,000 cycles at 20 A/g, demonstrating good stability. 

## 4. Conclusions

In summary, MnO_2_ nanoflakes with a thickness of ~1.2 nm and size of ~800 nm were obtained by ultrasonic exfoliation and charge regulated by the appropriate concentration of PDDA (0.75 g/L). Then MnO_2_ and GO nanosheets were self-assembled by electrostatic force. The composite of e-MnO_2_-0.75@rGO with layer-by-layer heterostructure was acquired after hydrothermal reduction by glucose. The composite exhibits excellent electrochemical performance. In 6 M KOH electrolyte, the specific capacitance of e-MnO_2_-0.75@rGO is 456 F/g at a current density of 1 A/g, which is much higher than that of pristine MnO_2_ (268 F/g). Even at 10 A/g, the specific capacitance still retains 201 F/g, and the specific capacitance retention is 98.7% after 10,000 charge-discharge cycles at 20 A/g. It shows that rGO improves the conductivity of the material, and the layer structure formed by rGO is conducive to the migration of ions in the electrolyte, so the specific capacitance of the composite is greatly enhanced. The improved electrochemical performance is attributed to the synergistic effect of architecture coupled with components, including the rate of ion transport and faradaic reaction, plenty of active sites, less restacking as well as improved electrical conductivity. Moreover, the assembled e-MnO_2_-0.75@rGO//GH ASC device shows a specific capacitance of 94 F/g at 1 A/g with a potential window of ~0–1.6 V and better cycle stability with capacitance retention of 99.3% over 10,000 cycles at 20 A/g. We believe that this work may provide a reference for the synthesis of the composites with layer-by-layer structure by self-assembly method and the basis for the design and comparison of electrode materials for high-performance supercapacitors. Reasonable design and construction of heterostructure have positive effects on the electrochemical performance. In future studies, we need to further optimize the structure and improve the electrochemical performance, as well as carry out researches on the interface mechanism of the heterostructure so as to better exploit the potential of MnO_2_/rGO composites in the application of supercapacitors.

## Figures and Tables

**Figure 1 membranes-12-01044-f001:**
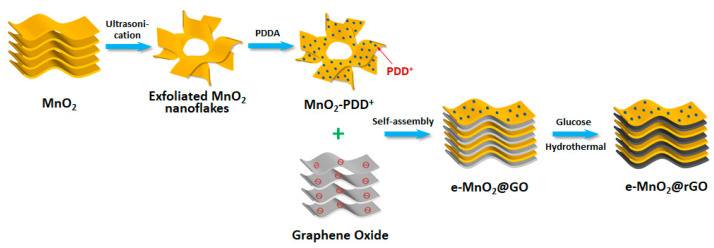
Schematic synthesis procedure of e-MnO_2_@rGO composites.

**Figure 2 membranes-12-01044-f002:**
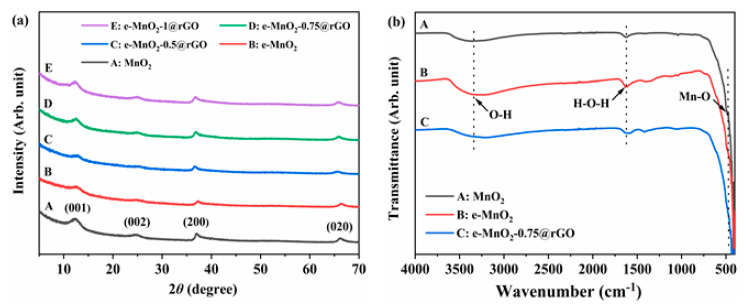
(**a**) XRD patterns of MnO_2_, e-MnO_2_, e-MnO_2_-0.5@rGO, e-MnO_2_-0.75@rGO and e-MnO_2_-1@rGO and (**b**) FTIR spectra of MnO_2_, e-MnO_2_ and e-MnO_2_-0.75@rGO.

**Figure 3 membranes-12-01044-f003:**
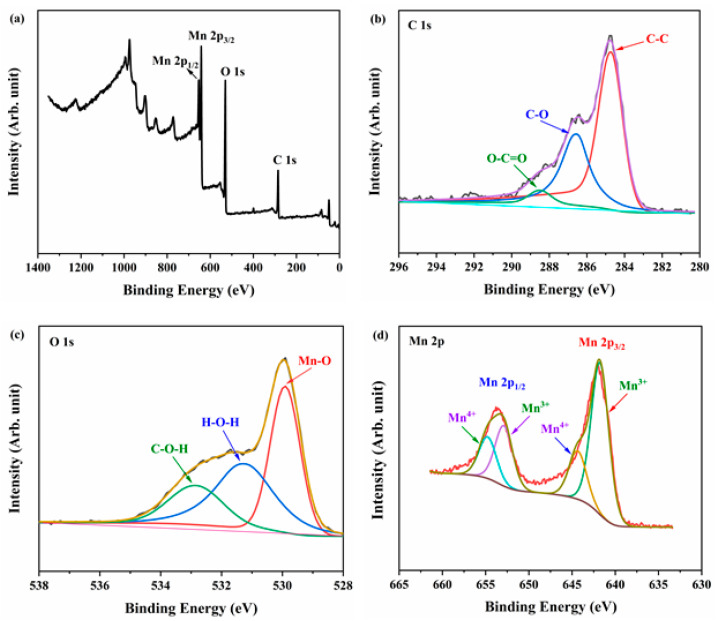
XPS spectra of (**a**) survey spectrum, (**b**) C 1s, (**c**) O 1s and (**d**) Mn 2p core level for e-MnO_2_-0.75@rGO.

**Figure 4 membranes-12-01044-f004:**
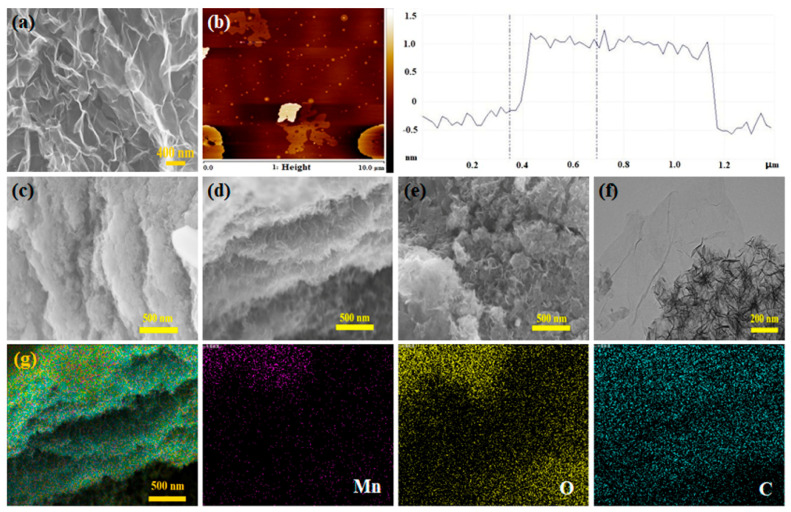
(**a**) SEM image of MnO_2_, (**b**) AFM image and the height profile of e-MnO_2_, (**c**) SEM image of e-MnO_2_-0.5@rGO (**d**) SEM image of e-MnO_2_-0.75@rGO, (**e**) SEM image of e-MnO_2_-1@rGO, (**f**) TEM image of e-MnO_2_-0.75@rGO and (**g**) SEM element mapping of e-MnO_2_-0.75@rGO.

**Figure 5 membranes-12-01044-f005:**
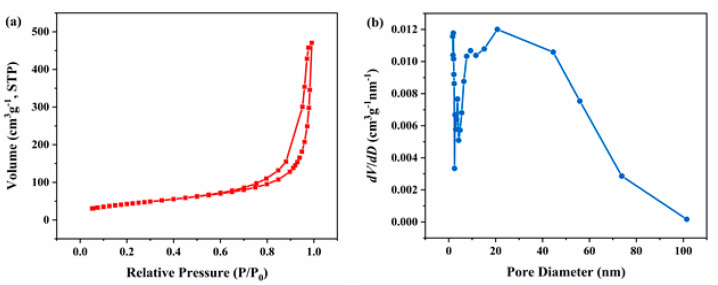
(**a**) Adsorption-desorption isotherms and (**b**) the pore-size distribution of e-MnO_2_-0.75@rGO.

**Figure 6 membranes-12-01044-f006:**
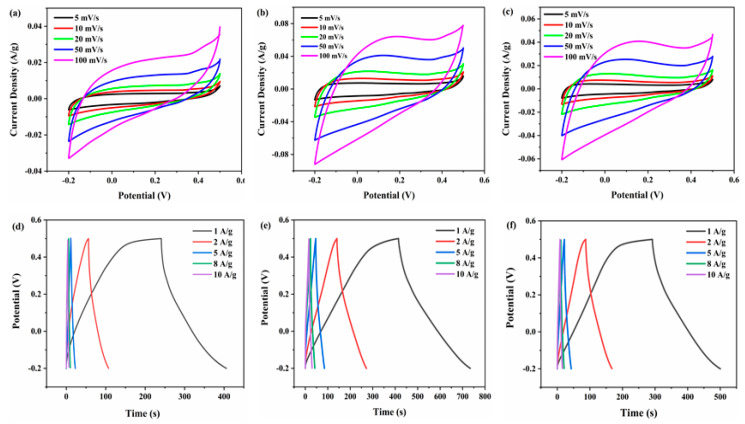
CV curves at various scan rates: (**a**) e-MnO_2_-0.5@rGO, (**b**) e-MnO_2_-0.75@rGO and (**c**) e-MnO_2_-1@rGO. GCD curves at various current densities: (**d**) e-MnO_2_-0.5@rGO, (**e**) e-MnO_2_-0.75@rGO and (**f**) e-MnO_2_-1@rGO.

**Figure 7 membranes-12-01044-f007:**
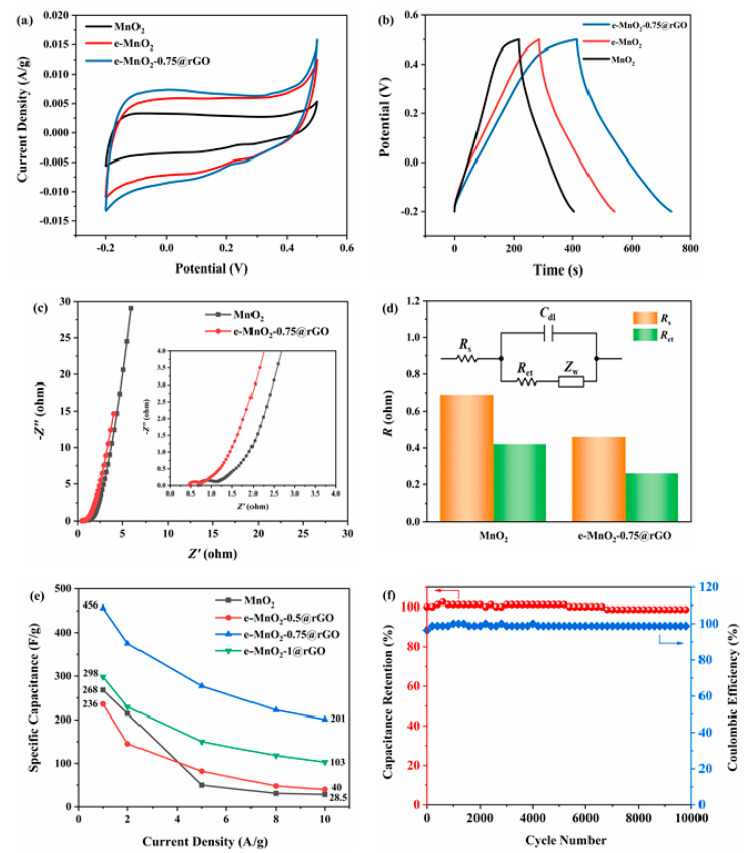
(**a**) CV curves (5 mV/s) of MnO_2_, e-MnO_2_ and e-MnO_2_-0.75@rGO, (**b**) GCD curves (1 A/g) of MnO_2_, e-MnO_2_ and e-MnO_2_-0.75@rGO, (**c**) Nyquist plots of EIS for MnO_2_ and e-MnO_2_-0.75@rGO, (**d**) Equivalent circuit and the values of fitting resistance, (**e**) Rate performance of MnO_2_, e-MnO_2_-0.5@rGO, e-MnO_2_-0.75@rGO and e-MnO_2_-1@rGO and (**f**) Cycle stability (20 A/g) and coulombic efficiency of e-MnO_2_-0.75@rGO.

**Figure 8 membranes-12-01044-f008:**
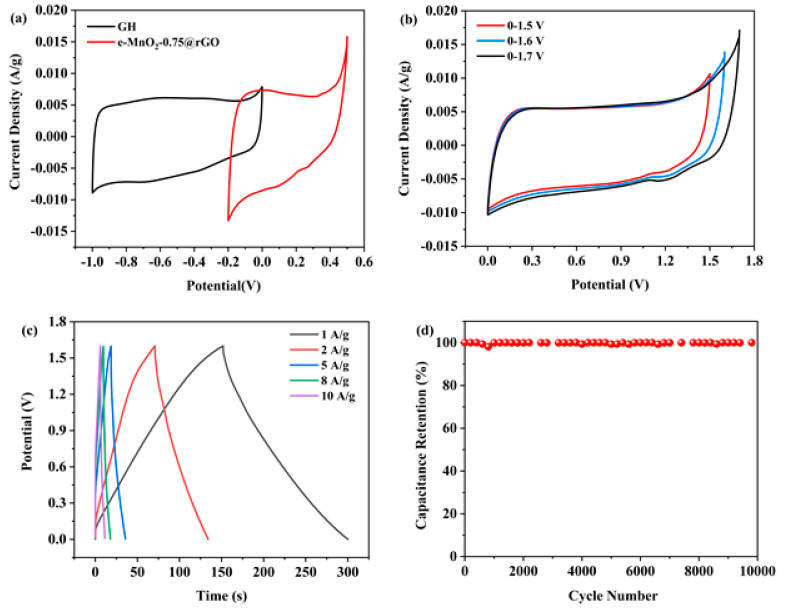
(**a**) CV curves of GH negative electrode and e-MnO_2_-0.75@rGO positive electrode (5 mV/s), (**b**) CV curves at different voltage windows (5 mV/s), (**c**) GCD curves at various current densities and (**d**) Cycle stability (20 A/g) of e-MnO_2_-0.75@rGO//GH ASC.

**Table 1 membranes-12-01044-t001:** A comparison of specific capacitance of e-MnO_2_-0.75@rGO in this work with other MnO_2_ composites in the previous literature.

Materials	Preparation Methods	Specific Capacitance	References
MnO_2_ NF/RGO@Ni foam	layer-by-layer (LBL) self-assembly	246 F/g (0.5 A/g)	[47]
*δ*-MnO_2_/modified graphene	Hydrothermal method	270 F/g (0.5 A/g)	[63]
Na-MnO_2_/rGO	Hydrothermal method	451 F/g (0.5 A/g)	[64]
rGO/C/MnO_2_	Carbonization + Hydrothermal treatment	215.2 F/g (0.15 A/g)	[65]
MnO_2_ and polyvinylpyrrolidone (PVP)@rGO	Electrodeposition	358 F/g (1 A/g)	[66]
MnO_2_/rGO	Sonochemical assisted synthesis	375 F/g (1 A/g)	[67]
MnO_2_/nitrogen-doped graphene (NG)	Hydrothermal method	305 F/g (5 mV/s)	[68]
MnO_2_/graphene	Hydrothermal method	255 F/g (0.5 A/g)	[69]
MnO_2_/rGO	Electrostatic self-assembly + Hydrothermal method	456 F/g (1 A/g)	This work

## Data Availability

Not applicable.

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
