# Peer review of "Layer-by-Layer Heterostructure of MnO2@Reduced Graphene Oxide Composites as High-Performance Electrodes for Supercapacitors"

_membranes, 2022, doi:10.3390/membranes12111044_

Round 1
Reviewer 1 Report (New Reviewer)
The authors report a fabrication procedure of layered MnO2 of a birnessite-type crystal structure with surface modification using polydiene dimethyl ammonium chloride. This material was further combined with graphene oxide, which was later reduced in a hydrothermal procedure. These materials were tested as supercapacitors. The manuscript is well written, with results well presented and full of interest for the Membrane readers, and should be published after minor comments.
1) Several authors report that the Mn 2p core level XPS spectra should exhibit multiplet splitting peaks related to contributions for two oxidation states, the Mn 3+ and Mn 4+. Please, check the following references to address the proper interpretation of it.
- Catalysts 2018, 8(4), 138; https://doi.org/10.3390/catal8040138
- IScience 2020, 23(1), 100797, https://doi.org/10.1016/j.isci.2019.100797
- RSC Advances 2021, 11(14), 7808, https://doi.org/10.1039/D0RA10376D
- Electrochimica Acta 2018, 261, 428,https://doi.org/10.1016/j.electacta.2017.12.118
Author Response
Please see the attachment.

Reviewer 2 Report (New Reviewer)
In this work, the authors reported the synthesis of the composite of MnO2@rGO and its application as electrode material for supercapacitors. The manuscript is well-written and easy to follow. The description of the method and characterization of the materials is very good, and I recommend its publication following minor revisions. Below is a short list of my comments and thoughts.
1. I am generally impressed with the paper's thoroughness. I wish a bit more of the ‘scientific’ thinking would shine through. This is probably more down to personal preference, but to me, it comes across as a bit sterile in writing. I wish there was a bit more reflection and conveying of knowledge (not just results) to the reader. Help the reader to make the right conclusions. For instance, I would recommend adding a short paragraph to the Discussion or Conclusion where you reflect on your findings and on how you may further improve the procedure.
2. “wavenumber” is one word (not Wave Number).
3. In science and technology, an arbitrary unit is abbreviated as Arb. unit (not a.u.).
4. Purity of using materials must be apparent.
5. Line 151, “constant temperature”, please clarify which temperature was used.
6. Please provide more details about the sample preparation for the FTIR measurements.
7. Why do the FTIR spectra demonstrate the spectral range from 4000 to 500 cm-1? According to the manufacturer, the FTIR spectrometer possesses a spectral range from 350 cm-1.
8. The band at 482 cm-1 was attributed to the stretching vibration of the Mn-O bond. However, in the FTIR spectra, this band is not visible. It would be correct to show the spectra in the lower region. Moreover, according to the literature the strong absorption band at 538 cm -1 can be allotted to the Mn-O stretching mode.
Author Response
Please see the attachment.

Reviewer 3 Report (New Reviewer)
The authors have prepared Layer-by-Layer Heterostructure of MnO2@Reduced Graphene Oxide Composites and studied several parameters that affect the High-Performance Electrodes for Supercapacitors. However, the manuscript needs major revision to be suitable for publication in the Journal.
Recommendation: It may eventually be publishable after revision but requires major revisions as indicated.
1) Abstract: Authors are given e-MnO2, without any clarification. Authors must explain.
2) Authors must explain more details in the introduction section about the “major advantages of reduced graphene oxide used in this application”
3) Figure quality is not good. Authors must check this part. In particular, Figure 3, Figure 5, and Figure 8.
4) Experimental Part: authors must check the experimental conditions carefully. Give a protocol on how to perform the assay, step by step. How many samples shall be taken? How shall it be treated? Which buffer and reagents (mg?; mL?; conc.?) must be added? How are analytical data generated? Specify analytical voltages. This is a major requirement for any method to be used by others. This is more important than stating where chemicals may have been purchased from.
5) Results: In the XRD analysis, both MnO2 and MnO2@rGO have the same peaks in Figure 2a. Unable to see rGO peak in the XRD. Why? Its major characterization for rGO in this research work. Authors must check again.
6) BET analysis: authors gave only a composite BET test report. Based on this data, which one influences the porosity aspect? MnO2 or rGO?, authors must explain clearly.
7) Results: EIS section, normally low Rct value of nanoparticles or nanomaterials are good for electrical connectivity. But here Au/g-C3N4/GCE is high. Is it good for the sensor working? Then only g-C3N4 is a high Rct value, why?
8) Authors developed asymmetric supercapacitor and chemical performance was evaluated. Authors must explain clearly the fabrication of the device (asymmetric supercapacitor).
9) Results: Some experimental data lack (relative) standard deviations. Give averaged data for important experimental data along with standard deviations (±) and the number of experiments (n =?).
Round 2
Reviewer 3 Report (New Reviewer)
Title: Layer-by-Layer Heterostructure of MnO2@Reduced Graphene Oxide Composites as High-Performance Electrodes for Supercapacitors
Journal; Membranes
In this work, authors developed Layer-by-Layer structure of MnO2@Reduced Graphene Oxide composite for energy applications. Current form of the manuscript is need for revision based on following commets.
1. In XRD analysis and Figure 2a: There is no difference between XRD peaks of MnO2 (A) and e-MnO2-1@rGO (C-D). Authors must explain, what is the difference between both, and how its confirm formation of composite?
2. The characteristic peak of (002) plane for rGO located at 25.1° [48], the position of which is close to (002) plane of δ-MnO2, so there may be overlap of (002) peaks for MnO2 and rGO, authors explain like "overlaping", its not reasonable. Authors better to add only rGO XRD analysis. in Figure 2a.
3. Also, reduced graphene oxide is not confirmed. Becuase, Figure 3a and XRD analysis, GO is reduced or not? Authors must check RGO XPS analysis.
4. Authors better to add BET analysis of RGO or MnO2 for comparsion analysis.
5. Better to do device fabrication and performances using this composite. If you have any data related with device, add in the revised manuscript.
6. Authors better to improve introduction section more effectively based on advantages of binary sheets and RGO in the supercapcitors application with following references (a). https://doi.org/10.1039/C5TA10297A
(b). https://www.sciencedirect.com/science/article/pii/S0925838822039445?via%3Dihub
(c). https://www.sciencedirect.com/science/article/pii/S0013468618322904
(d). https://pubs.acs.org/doi/full/10.1021/am4003843
(7). There are some grammatical and typographical errors. Please check the manuscript and refine carefully.
(8). Comparison table should improve it. Its highly helpful for reader of the articles and researchers.
Author Response
Please see the attachment.

This manuscript is a resubmission of an earlier submission. The following is a list of the peer review reports and author responses from that submission.
Round 1
Reviewer 1 Report
This work reports the preparation of MnO2 nanoflakes by ultrasonic exfoliation, and self-assembled with rGO nanosheets by electrostatic force with layer-by-layer heterostructure, acquired after hydrothermal reduction by glucose. One of the synthesized MnO2/rGO composites exhibits remarkable electrochemical performance. The work is raised through a correct scientific reasoning.
However, the main weakness of the manuscript is the lack of explanation about the novelty or improvement it brings. In the literature on electrodes for supercapacitors, a large number of articles reporting the preparation of nanostructured MnO2 on carbon materials are known, presenting spectacular results. For the article to be considered as novel, the articles should carry out a comparative study with the previous bibliography, especially with regard to MnO2-rGO composites for supercapacitors (several papers); and more specifically, an in-depth analysis should be carried out comparing the strategy and results presented here with those previously reported by other authors for LbL MnO2-graphene based composites.
Very similar previous articles such as 10.1016/j.jpowsour.2016.11.096; 10.1016/j.tsf.2020.138483; 10.1016/j.ceramint.2016.10.032; 10.1016/j.electacta.2017.12.071; …. and many more similar ones should be compared to justify the novelty and relevance of the work presented.
Reviewer 2 Report
In this paper, δ-MnO2 with layered structure was prepared by a facile liquid phase method, and the MnO2 nanosheet was obtained by ultrasonic exfoliation. Then, positive charges were grafted on the surface of MnO2 nanosheets with polycation electrolyte of polydiallyl dimethylammonium chloride (PDDA) in different concentrations. Series composites of e-MnO2@reduced graphene oxide (rGO) were obtained by electrostatic self-assembly combined with hydrothermal chemical reduction. The work of this paper is excellent, but there are still some issues that need to be pointed out.
1. What are the advantages of using PDDA for the author of this article, and whether there are other compounds that have the same effect as PDDA, please make some additional clarifications.
2. One sub-section to introduce the raw materials should be provided.
3. Fig. 1 is suggested to shift into experimental section.
4. What is the meaning of the graph on the right of Figure 4b?
5. The role of the reduce graphene oxide for the electrodes should be further clarified with supporting articles: Energy Technology 8 (9), 2000397, 2020; Nitrogen, sulfur co-doped hierarchical carbon encapsulated in graphene with “sphere-in-layer” interconnection for high-performance supercapacitor; etc.
6. It is suggested to combine the results section and discussion section together. The corresponding results should be discussed one by one.
7. It is suggested that the authors add the specific capacitance data of e-MnO2-0.5@rGO, e-MnO2-0.75@rGO and e-MnO2-1@rGO to the rate performance graph (Figure 7e).
8. MnO2 is a common material for the electrodes. More background on the advantages of MnO2 and its applications in supercapacitor electrodes should be provided with supporting articles: Wood‐Derived High‐Mass‐Loading MnO2 Composite Carbon Electrode Enabling High Energy Density and High‐Rate Supercapacitor; Polymer 235, 124276, 2021; etc. In addition, more comparison to the capacitance performance with previous reports should be provided in a table or a Ashby plot.
9. Where is the Table 1?
10. This article has some grammatical errors and vocabulary usage issues, please make corrections.